# Effects of climate and forest development on habitat specialization and biodiversity in Central European mountain forests
Tobias Richter [1,2] ✉, Lisa Geres[1,2,3], Sebastian König [1,2], Kristin H. Braziunas [1], Cornelius Senf [4], Dominik Thom [1,5,6], Claus Bässler[7,8], Jörg Müller [8,9], Rupert Seidl [1,2] & Sebastian Seibold [1,2,10]

Mountain forests are biodiversity hotspots with competing hypotheses proposed to explain elevational trends in habitat specialization and species richness. The *altitudinal-niche-breadth hypothesis* suggests decreasing specialization with elevation, which could lead to decreasing species richness and weaker differences in species richness and beta diversity among habitat types with increasing elevation. Testing these predictions for bacteria, fungi, plants, arthropods, and vertebrates, we found decreasing habitat specialization (represented by forest developmental stages) with elevation in mountain forests of the Northern Alps – supporting the *altitudinal-niche-breadth hypothesis*. Species richness decreased with elevation only for arthropods, whereas changes in beta diversity varied among taxa. Along the forest developmental gradient, species richness mainly followed a U-shaped pattern which remained stable along elevation. This highlights the importance of early and late developmental stages for biodiversity and indicates that climate change may alter community composition not only through distributional shifts along elevation but also across forest developmental stages.

Mountain regions are hotspots of biodiversity globally and characterized by strong changes in biodiversity along elevational gradients. Various hypotheses have been proposed to describe why species richness commonly decreases with increasing elevation[1–3]. Environmental filtering theory, for example, predicts that only species with certain adaptations to harsh climatic conditions are able to persist at high elevations[4–6]. The *altitudinal-niche-breadth hypothesis* provides more detail by suggesting that due to increasing spatial and temporal environmental variability with increasing elevation, species at high elevation exhibit stronger population fluctuations and are less specialized with regard to, for example, the available resources or the prevailing biotic interactions[7]. Lower specialization allows species to persist under more variable environmental conditions but restricts resource partitioning, with the result that fewer species are able to coexist[8–11]. Some studies provide evidence supporting the *altitudinal-niche-breadth*

*hypothesis*[7,12,13] and a recent meta-analysis confirmed a positive correlation between community-level specialization and species richness[14], yet the generality of these mechanisms across trophic levels and taxonomic groups remains inconclusive.

In forest ecosystems, pulses of tree mortality are a major driver of ecosystem dynamics. Tree senescence and disturbance open the forest canopy, reset forest development, and create temporal and spatial variability in abiotic and biotic conditions[15,16]. After the disturbance, forests typically undergo a sequence of developmental stages characterized by distinct differences in forest structure[17]. Many forest species are specialized in habitat conditions provided by different developmental stages[18]. Thus, forest-dwelling species diversity and composition vary along forest development[19], with high diversity in early and late developmental stages, and low diversity in intermediate developmental stages[20]. Yet, it remains unclear whether

[1]Technical University of Munich, School of Life Sciences, Ecosystem Dynamics and Forest Management Group, Hans-Carl-von-Carlowitz-Platz 2, Freising, Germany. [2]Berchtesgaden National Park, Doktorberg 6, Berchtesgaden, Germany. [3]Goethe University Frankfurt, Faculty of Biological Sciences, Institute for Ecology, Evolution and Diversity, Conservation Biology, Frankfurt am Main, Germany. [4]Technical University of Munich, School of Life Sciences, Earth Observation for Ecosystem Management, Hans-Carl-von-Carlowitz-Platz 2, Freising, Germany. [5]Chair of Silviculture, Institute of Silviculture and Forest Protection, TUD Dresden University of Technology, Pienner Str. 8, Tharandt, Germany. [6]Gund Institute for Environment, University of Vermont, 617 Main Street, Burlington, VT, USA. [7]Ecology of Fungi, Bayreuth Center of Ecology and Environmental Research (BayCEER), University of Bayreuth, Universitätsstr. 30, Bayreuth, Germany. [8]Bavarian Forest National Park, Freyunger Strasse 2, Grafenau, Germany. [9]Ecological Field Station Fabrikschleichach, Department of Animal Ecology and Tropical Biology, University of Würzburg, Glashüttenstraße 5, Rauhenebrach, Germany. [10]Forest Zoology, Institute for Forest Botany and Forest Zoology, TUD Dresden University of Technology, Pienner Str. 7, Tharandt, Germany. ✉e-mail: tobias.richter@npv-bgd.bayern.de

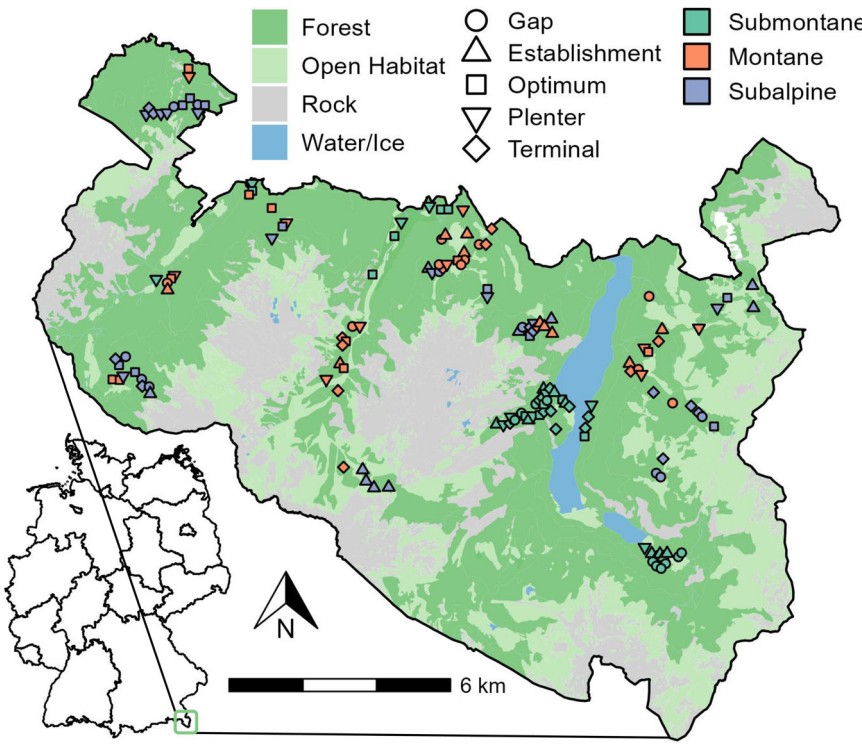

**Fig. 1 | Study area.** Berchtesgaden National Park and its location in Germany. The map shows the main habitats (forest, open habitats, rock, water/ice) based on data provided by the National Park administration. Plot coordinates were measured using a Trimble r12i GNSS receiver during field sampling. Point shapes illustrate the forest developmental stage and point colours the elevational zone.

diversity patterns along forest development are modulated by climatic conditions, for example along elevational gradients. Specifically, the interactive effects of disturbance and the processes driving the *altitudinal-niche-breadth hypothesis* have not been investigated to date. Given that climate is warming more strongly in mountain areas[21] and disturbance regimes are changing rapidly in many mountain landscapes[22,23], understanding how these changes – and the resultant prevalence of different developmental stages in the landscape – affect forest-dwelling species is crucial to better predict the impacts of climate change on biodiversity.

To test the interactive effect of elevation and forest development after disturbance on habitat specialization and biodiversity, we set up a network of 150 study plots covering the full gradient of forest development replicated ten times each across three elevational zones (submontane, montane, subalpine) in the Northern Alps (Fig. 1). At each plot, we sampled multi-trophic diversity – covering bacteria, fungi, plants, arthropods, and vertebrates – and assessed habitat specialization (i.e., a species´ proportional use of different forest developmental stages), species richness, and beta diversity along forest developmental stages and elevation, respectively. Based on the processes underlying the *altitudinal-niche-breadth hypothesis* and considering a positive specialization-species richness correlation, we predict that both (H I) habitat specialization and (H II) species richness decrease with increasing elevation (Fig. 2 I & II). Given decreasing specialization with elevation, we further predict that (H III) community dissimilarity across developmental stages is highest at the submontane and lowest at the subalpine zone (Fig. 2 III). Combining both the elevational and the forest developmental gradient, we finally predict that (H IV) differences in species richness among forest developmental stages are strongest at lower elevations and decrease towards the tree line (Fig. 2 IV).

## Results

In total, we recorded 12,687 and 9,173 Operational Taxonomic Units (OTUs, see methods) of bacteria and fungi, respectively, 443 plant species, 958 arthropod species identified by taxonomists (termed arthropods$_{TAX}$), 8,335 Barcode Index Numbers (BINs, see methods) of insects identified by DNA-metabarcoding (termed insects$_{BIN}$), and 105

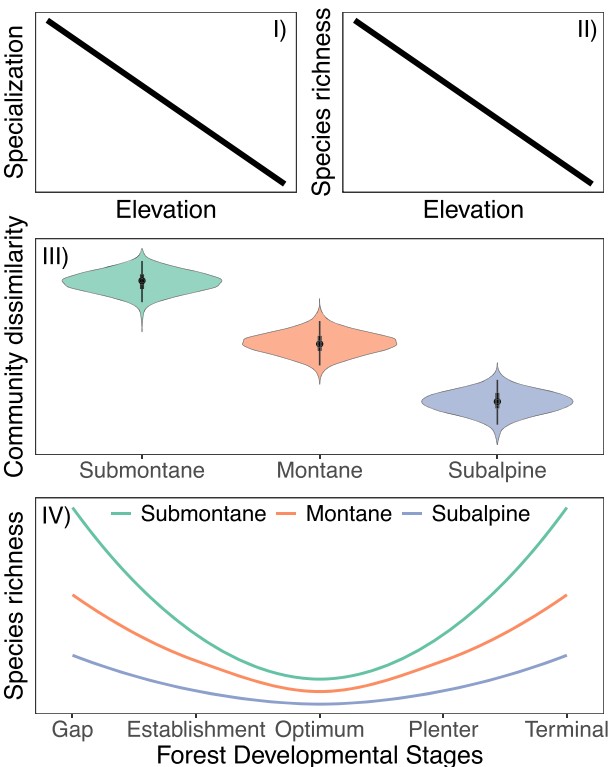

**Fig. 2 | Graphical concept.** Based on the *altitudinal-niche-breadth* hypothesis and a positive specialization-species richness correlation, we expect both decreasing (H I) habitat specialization and (H II) species richness with elevation. Given decreasing specialization with elevation, we expect (H III) decreasing community dissimilarity across developmental stages with elevation. Given decreasing community dissimilarity with elevation, we finally predict that (H IV) differences in species richness among forest developmental stages are strongest at lower elevations and decrease towards the tree line.

vertebrate species (Supplementary Table S3). On average across all plots, we found a very broad habitat use ranging between 3.6 developmental stages used by fungi and 4.5 stages used by vertebrates. Vertebrates showed the lowest species richness (28 species) and lowest Jaccard dissimilarity (0.54), whereas bacteria showed the highest species richness (1382 OTUs) and fungi the highest Jaccard dissimilarity (0.84). All descriptive statistics are shown in the Supplementary Table S3.

## Patterns of habitat specialization across elevational zones

Using the standardized effect size (SES) from a null model that allows to account for the increasing probability of using more forest developmental stages with more occurrences (supplementary Fig. S8), we found strong evidence of higher habitat specialization in the submontane compared to the montane and subalpine zone for all taxa (Fig. 3), as indicated by the probability of direction (pd ≥ 0.99, a Bayesian measure of effect existence) and the ROPE percentage (0% in ROPE, a Bayesian measure of effect importance, Supplementary Table S5; both measures described in more detail in the methods). Between the montane and subalpine zones, patterns of specialization differed among taxa. Arthropods_TAX showed a lower specialization in the subalpine compared to the montane zone (pd = 1.00, 0% in ROPE). Bacteria, fungi, and plants showed a higher specialization in the subalpine compared to the montane zone (pd = 1.00). This difference was only of substantial magnitude for bacteria and plants (0% in ROPE) and not for fungi (4% in ROPE). We found no evidence for a difference in habitat specialization between the montane and subalpine zones for insects_BIN and vertebrates (pd ≤ 0.69, ≥31% in ROPE).

## Patterns of species richness along elevation

Arthropod_TAX and insect_BIN richness decreased by 16.1% [95% HDI_low = 11.7, 95% HDI_high = 20.4] and 10.2% [4.1, 14.9], respectively, with one unit of SD of elevation (=359 m) (Fig. 3, Supplementary Table S7). The decrease is well supported for both taxa (pd = 1.00) and was substantial for arthropods_TAX (0% in ROPE) but small for insects_BIN (43% in ROPE). Species richness of bacteria and fungi decreased with one unit SD of elevation by 0.7% [−5.2, 4.3] and 1.4% [−7.4, 5.8], and species richness of plants and vertebrates increased by 4.3% [−1.4, 12.8] and 3.0% [−1.8, 9.3], respectively. These changes can be seen as uncertain and negligible (pd < 0.94, >94% in ROPE).

## Elevational patterns of beta diversity among developmental stages

All taxa except arthropods_TAX showed the highest Jaccard dissimilarity among forest developmental stages either in the submontane zone (insects_BIN and vertebrates) or Jaccard dissimilarity was equally high in both the submontane and subalpine zone (Fig. 4). For insects_BIN and vertebrates, Jaccard dissimilarity was lower in the montane and subalpine compared to the submontane zone, well supported for vertebrates (pd = 1.00, 0% in ROPE) but not for insects_BIN (pd ≤ 0.95, ≥3% in ROPE) (Supplementary Table S11 & Fig. S13). For bacteria and plants, Jaccard dissimilarity was lowest in the montane (pd = 1.00, 0% in ROPE) but did not differ between the submontane and subalpine zone (pd < 0.66, 25% in ROPE). Jaccard dissimilarity of fungi did not change across the elevational zones (pd < 0.73, >7% in ROPE). Jaccard dissimilarity of arthropods_TAX was lowest in the submontane (pd = 1.00, 0% in ROPE), but did not differ between the montane and subalpine zone (pd = 0.82, 11% in ROPE).

## Interactive effects of forest development and elevation on species richness

Species richness of all taxa except bacteria followed a U-shaped pattern along the forest developmental gradient (Fig. 5). Fungi, plant and vertebrate species richness was about 23% [7, 32], 18% [6, 32] and 8% [-4, 17] lower in intermediate stages compared to gaps, on average across all elevations (Supplementary Tables S8a–S8c). While for both fungi and plants lower species richness in intermediate stages was strongly supported (pd = 1.00), the decrease was substantial only for fungi (<3% in ROPE) and of moderate magnitude for plants (<7% in ROPE). Lower vertebrate species richness in intermediate developmental stages was uncertain and negligible (pd = 0.89, >72% in ROPE). Insect_BIN species richness was substantially higher (between 19% [10, 27] and 28% [21, 36]) in gaps compared to all other stages (pd = 1.00, 0% in ROPE). Species richness of arthropods_TAX was 9.9% [1, 17] higher in gaps and 12.8% [4, 20] higher in the terminal stage compared to the establishment stage. Bacteria showed no notable differences (between 1.2% [−8, 12] and 4.9% [−5, 16]) in species richness between gaps and all other developmental stages (pd < 0.81, >87% in ROPE).

We found no substantial differences in species richness patterns among developmental stages in lower compared to higher elevations for any taxonomic group (all pd ≤ 0.91, always >3% in ROPE; Supplementary Table S9 & Fig. S11). Patterns in Fig. 6 suggest substantial differences between individual developmental stages, but credible intervals were wide

---

**Fig. 3 | Habitat specialization and species richness along elevation.** Standardized effect size (SES) of habitat specialization and normalized species richness along the elevational gradient. We calculated habitat specialization using the reciprocal of the Simpson index, based on the number of different forest developmental stages used by a species within each elevational zone. We then used a null model which allows to account for the increasing probability of using more developmental stages with increasing occupancies (Supplementary Fig. S7), calculated an SES, and averaged across all species per plot for each elevational zone (see methods for more details). Predictions are from individual Bayesian multilevel models for each measure and taxon. We averaged the predictions across forest developmental stages and summarized by means of the MAP and 95% HDI (explanation in methods). To show all taxa in a joint figure for species richness, we normalized the predictions of species richness to a range between zero and one based on each taxon's minimum and maximum values (see Table S3 and Fig. S9 in the supplement for values and figures including data points).

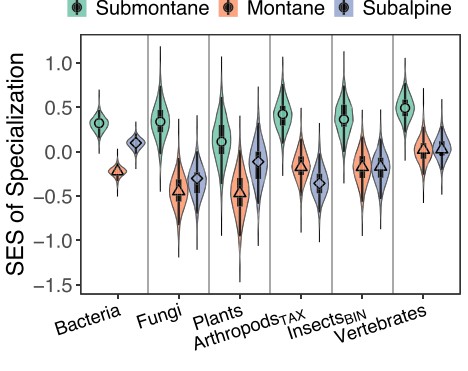

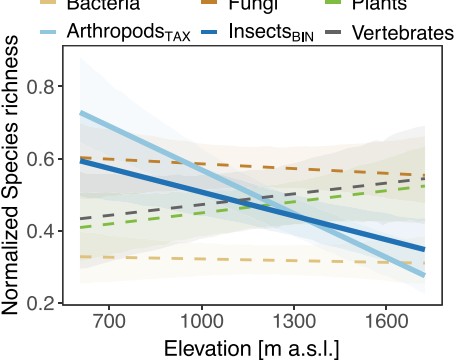

**Fig. 4 | Jaccard dissimilarity across elevational zones.** Jaccard dissimilarity for each taxonomic group across the three elevational zones. Jaccard dissimilarity was calculated between each plot pair of a different developmental stage within each elevational zone, modelled and predicted using individual Bayesian multilevel models with a beta probability distribution, separately for each taxonomic group. Posterior distributions were summarized by means of the MAP and 95% and 50% (thick bars) HDI (explanation in methods). Different point shapes indicate substantial differences in Jaccard dissimilarity between elevational zones. Supplementary Fig. S13 includes data points.

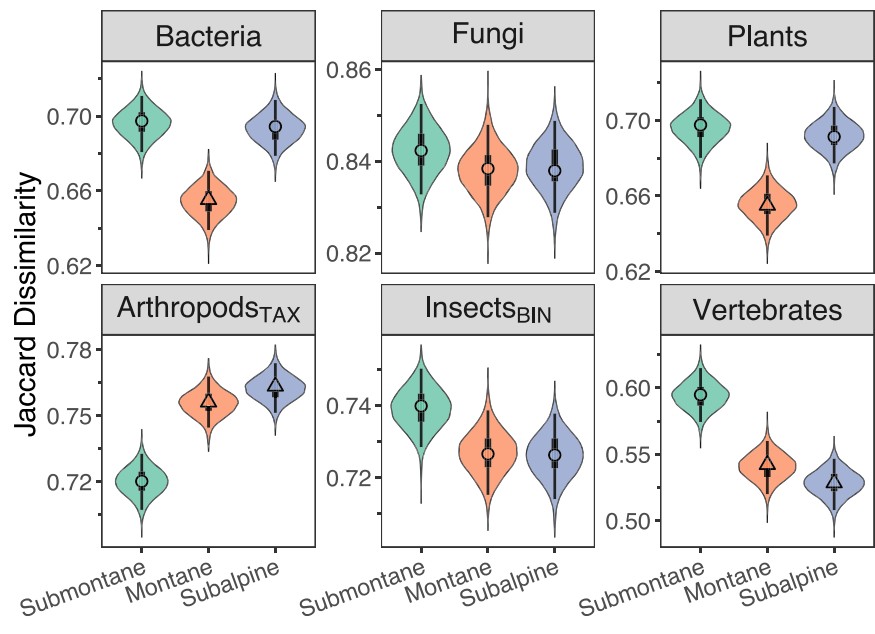

**Fig. 5 | Species richness along forest development.** Patterns of species richness along forest development (sensu Zenner et al.[63]) for each taxonomic group from individual Bayesian multilevel models with a negative binomial error distribution. We predicted 50 times along the elevational gradient, averaged them by developmental stage, summarized the means using the MAP (explanation in methods), normalized them between the minimum and maximum predicted value of each taxon, and fitted a loess curve for better visualization.

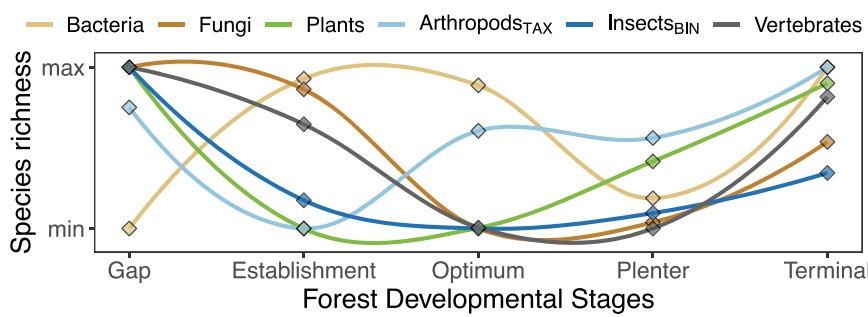

**Fig. 6 | Species richness along forest development and elevation.** Patterns of species richness along forest development (sensu Zenner et al.[63]) across elevational zones for each taxonomic group. We predicted species richness for each developmental stage at 719 m, 1154 m, and 1565 m, which are approximately the central elevations of each elevational zone, using Bayesian multilevel models with a negative binomial error distribution. We summarized the predictions using the MAP (explanation in methods) and fitted a loess curve for better visualization. The patterns suggest substantial differences among elevational zones, but high data variability resulted in wide and overlapping credible intervals (Supplementary Fig. S11 includes data points and credible intervals), leading to only marginal differences (Supplementary Table S9 & Fig. S10).

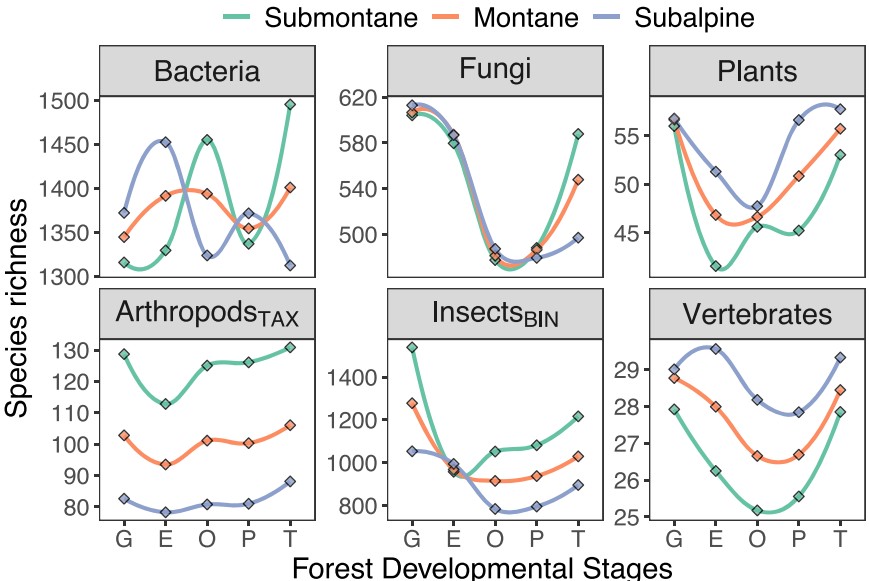

and largely overlapped (Fig. S12). While patterns of all taxa varied little across elevational zones, bacteria showed opposing patterns in the submontane and subalpine zone (Fig. 6).

## Discussion

Our study provides strong evidence for the *altitudinal-niche-breadth hypothesis*. In particular, we found decreasing habitat specialization with elevation for all six taxa investigated. Yet, species richness decreased with elevation only for two taxa. Species richness differed among forest developmental stages in all taxa and showed mainly a U-shaped pattern along the stages, which remained stable along elevation. Beta diversity among the stages, however, varied across elevational zones for most taxa but aligned with the elevational patterns of habitat specialization. This suggests that climate and forest disturbances (i.e., the causal driver of forest developmental stages) interactively affect species communities in Central European mountain forests.

Confirming our hypothesis (H I) and in line with the *altitudinal-niche-breadth hypothesis*, habitat specialization decreased with elevation for all six taxa. As a mechanism behind this decrease, the *altitudinal-niche-breadth hypothesis* proposes decreasing specialization with elevation due to harsher and more variable environmental conditions[7,24]. This mechanism might explain the observed patterns in our study, since both decreasing temperature and increasing microclimatic variability are in line with this prediction (Supplementary Fig. S2c). The increase in microclimatic variability with elevation in mountain forests is partly driven by decreasing and more variable canopy cover, resulting in reduced thermal buffering and more heterogenous and extreme microclimates[25,26] (Supplementary Fig. S2a & b). While specialists may struggle to survive under such fluctuating conditions, generalists might be able to make use of a wider range of microclimatic conditions or resources[7]. Moreover, although microclimatic variability was higher in all forest developmental stages in high-elevation forests, differences in canopy cover and microclimatic variability among developmental stages were less pronounced in the subalpine than in lower elevational zones (Supplementary Fig. S2). This could also allow more species of high-elevation forests to occur in different developmental stages. However, we observed for all taxa except arthropods_TAX that specialization did not change or even increased from the montane to the subalpine zone. One possible explanation is that topographic complexity and variability in soils under a more open and variable canopy cover increases habitat heterogeneity at higher elevations, which might promote habitat specialists by weakening microclimatic variability through a fine-scaled mosaic of microhabitats[27–32]. Moreover, it is important to note that specialization is not only affected by abiotic drivers, but also by associations between taxa[7]. This may explain similar patterns in habitat specialization among bacteria, fungi, and plants in our study since there are various associations between species of these groups[33–37]. Finally, specialization may not only drive species richness, but species richness could also affect specialization, as suggested by a recent meta-analysis on latitudinal specialization-richness patterns[14], which could be a further explanation for patterns (partly) inconsistent with the *altitudinal-niche-breadth hypothesis*[7].

Decreases in species richness along elevational gradients have been previously reported[1–3,38,39] and can be caused by different mechanisms[2]. Considering a positive correlation between specialization and species richness[14] in combination with the assumptions made by the *altitudinal-niche-breadth hypothesis*[7], we expected decreasing species richness with elevation (H II). We found patterns consistent with our hypothesis for arthropods_TAX and insects_BIN. This suggests that at least for arthropods_TAX and insects_BIN, lower specialization at higher elevations could be one mechanism behind decreasing species richness with increasing elevation. As a further mechanism, environmental filtering theory suggests that harsh climatic conditions beyond the thermal limits of many species can restrict the number of species able to persist at higher elevations[40–42], and thus lower temperatures are often associated with lower species richness[39,43]. These mechanisms could have contributed to the elevational richness patterns of insects and other arthropods since these are ectothermic species[4,40,44,45].

Environmental filtering could also be the mechanism behind the slight increase in plant species richness with elevation, but with light availability being more important than temperature. Due to overall lower canopy cover, light (energy) availability in the understory is higher in high- than low-elevation forests[26] (Supplementary Fig. S2a). High light availability promotes a higher plant cover, consequently allowing for a greater number of species with viable populations – in line with the *more-individuals hypothesis*[46–48]. This implies that for plants, environmental filtering may be stronger at lower elevations, limiting the number of species living beneath closed canopies and leading to higher specialization. Species richness of soil bacteria and fungi declined weakly with elevation, despite a clear decrease in specialization. This weak link between specialization and species richness of soil bacteria and fungi could be explained by the generally strong dependence of these taxa on soil pH, nutrient availability, and plant species[33–35], interfering with general mechanisms shaping species richness along elevational gradients. The marginally increasing species richness of vertebrates with elevation was mainly driven by bats and may also be explained by canopy openness of high-elevation forests. More open forests allow more bat species to forage in more developmental stages at higher elevations[49,50]. Our findings emphasize the crucial role of canopy cover in driving habitat specialization and species richness patterns in Central European mountain forests. Yet, taxa are affected differently by factors that do not change uniformly along the elevational gradient, leading to deviations from the expected positive specialization–richness relationship.

Considering that specialization was predicted to decrease with elevation, we hypothesized that beta diversity between forest developmental stages also decreases with elevation (H III). This hypothesis was supported for the majority of taxa (bacteria, plants, insects_BIN, and vertebrates), showing congruent patterns with specialization and strongest changes from the submontane to the montane zone. This indicates that decreasing habitat specialization with increasing elevation leads to taxonomic homogenization across forest developmental stages for these taxa[51]. However, higher similarity in forest structure and microclimate among developmental stages (despite increasing overall variability in microclimate with elevation; Supplementary Fig. S2) may have contributed to the observed pattern. Stabilizing or increasing beta diversity from the montane to the subalpine zone, similar to habitat specialization, may result from greater within-stage habitat heterogeneity, promoting species turnover and admixture of open habitat species[27–32]. Patterns of fungi and arthropods_TAX were inconsistent with our hypothesis. Even though specialization decreased for both, beta diversity patterns remained constant or increased, indicating stronger clustering with elevation. This suggests that developmental stages determine habitat conditions at lower elevations while other factors, such as topographic complexity or resource availability, may become more important at higher elevations. Furthermore, decreasing occurrences and a higher proportion of rare arthropod_TAX species with increasing elevation may have caused greater differences in community composition among developmental stages (Supplementary Table S4). Overall, differences of beta diversity among developmental stages at different elevational zones suggest that forest development and climate interactively shape forest community compositions.

Species richness changes after disturbances[52–54] and many taxa show U-shaped species richness patterns along forest developmental gradients[20]. We found support for a U-shaped pattern of species richness for all taxa except bacteria – highlighting the importance of early and late developmental stages for forest biodiversity[15,55–57]. However, contrary to our expectations (H IV), differences in species richness among developmental stages were rather stable and not generally weaker at higher elevations. We assumed the opposite due to lower canopy cover and higher microclimatic similarity among developmental stages at high elevations (Supplementary Fig. S2). However, high topographic complexity in mountain landscapes probably weakens or masks the macroclimatic effects of elevation[27–32]. Furthermore, high habitat heterogeneity provides habitats for a large number of species including many rare species[58–60], which are, however, more frequently subject to stochastic processes[61,62]. This may explain why

differences in species richness among forest developmental stages did not differ clearly between elevational zones as indicated by the wide credible intervals observed.

Our study shows that elevation and post-disturbance forest development are major drivers of biodiversity across a wide range of forest-dwelling taxa in a mountain forest of the European Alps. Observed patterns of habitat specialization aligned with the *altitudinal-niche-breadth hypothesis* for all taxa and were largely congruent with beta diversity. Yet, the expected decline in species richness with elevation was observed only in insects and other arthropods. This indicates that the mechanisms behind the *altitudinal-niche-breadth hypothesis* do play a role across all trophic levels in our study system, but species richness patterns are not generally explained by positive specialization-richness relationships. In contrast to species richness patterns along forest development, beta diversity among developmental stages varied with increasing elevation for most taxa. This suggests that climate (represented by the elevational gradient) and disturbance (represented by the forest developmental gradient) independently drive patterns of species richness, but interactively shape community composition.

We assessed habitat specialization based on forest developmental stages[63], which integrate multiple environmental conditions instead of measuring them individually. Since it is impractical to measure all relevant environmental variables across a wide range of taxa, using forest developmental stages as a proxy offers a more general picture of habitat differences. However, it does not allow to link patterns of species diversity to individual environmental drivers. Nevertheless, forest developmental stages are a widely applied concept in forest ecology and help guide management decisions[17,63–66]. For testing hypotheses related to elevational patterns in ecology, ideally the full elevational gradient is covered. In this study, the elevational gradient reached the tree line but was truncated at the lower end at approximately 600 m asl. Extending the gradient to lower elevations, however, would have introduced considerable bias due to differences in forest management and fragmentation[67–69], as well as by orders of magnitude larger spatial distances between study plots. We are confident that the observed elevational patterns would be similar if elevations below 600 m had been included since the submontane zone has a similar tree species composition and forest development regime as forests below 600 m[70].

Climate change is leading to altered climatic conditions and forest dynamics[71–73], which will have far-reaching effects on biodiversity[74]. If low elevations are a proxy for future climate at higher elevations, our results suggest that in our study system species richness of insects and other arthropods will increase, whereas species richness of plants and vertebrates will decrease as a result of a warming climate. Our results also suggest that differences in species richness among forest developmental stages will remain stable, but differences in community composition will be more pronounced under a warming climate. If increasing forest disturbances result in a greater proportion of the landscape at early developmental stages[73], our results suggest positive effects on species richness for most taxonomic groups. This underscores the potential of early developmental stages without human intervention, such as salvage logging, for promoting species richness.

## Methods
### Study area
This study was conducted at Berchtesgaden National Park located in the northern limestone Alps of south-east Germany (Fig. 1). The study area is characterized by high topographic complexity and environmental heterogeneity due to the steep terrain, with elevation ranging from 603 m (lake Königssee) to 2713 m asl (Mt. Watzmann). Forests cover approximately 54% of the national park's area of roughly 21,000 ha, with the tree line at approximately 1700 m asl[75]. The natural tree species composition differs between elevational zones: European beech (*Fagus sylvatica* L.) dominates the submontane zone (<850 m asl). Mixed forests consisting of European beech, Norway spruce (*Picea abies* (L.) Karst.) and Silver fir (*Abies alba* Mill.) are common in the montane zone (850–1400 m asl), and conifer forests of Norway spruce, European larch (*Larix decidua*) and Swiss stone

pine (*Pinus cembra*) dominate the subalpine zone (1400–1700 m asl)[70,76]. The region has a long history of timber extraction for salt mining, which has led to increased shares of Norway spruce in the submontane and montane zone[77]. Canopy cover decreases and variation in canopy cover increases with elevation[26] (Supplementary Figs. S2a & S2b), lowering the thermal buffering capacity[25,78] and leading to relatively higher microclimatic variability at intermediate to late developmental stages but lower variation across stages in the subalpine zone (Supplementary Fig. S2c). Conventional forest management ceased when the national park was founded in 1978, and is today restricted to restoration management carried out on 25% of the park, restoring the natural species composition by planting European beech and Silver fir[79] and preventing the spread of bark beetle outbreaks to neighbouring commercial forests.

### Classification of forest developmental stages and plot selection
We modified a classification protocol[63] to select plots covering the full gradient of forest development (Supplementary Fig. S3), differentiating five forest developmental stages: gap, establishment, optimum, plenter, terminal. We first applied the protocol to data of the last available forest inventory (2010-2012) to pre-select approximately 300 candidate plots. The candidate plots covered all major forested areas of the national park, except areas where bark-beetle trees are felled and those that were too steep to be accessed. We visited all candidate plots in the field in 2020 to verify and adjust the pre-classified developmental stage, and to exclude plots that intersected with hiking trails. We finally selected ten replicates per forest developmental stage and elevational zone (for ranges see study area), resulting in 150 plots in total (Fig. 1). Submontane plots were more clustered compared to plots in the other elevational zones, as only a few areas in the park are below 850 m asl and thus in the submontane zone (Fig. 1). The plots were circular and covered an area of 500 m² ($r = 12.62$ m) with a minimum distance between plot centres of 125 m. The lowest plot was located at 605 m and the highest at 1725 m asl. Figure S4 in the supplementary material shows a compilation of selected plots.

### Field sampling and species identification
We collected data from 14 taxonomic groups across all 150 plots (Table 1). Field sampling was conducted in 2021 except light trapping, which was conducted in 2022. Bacterial and fungal communities in the soil were identified through DNA metabarcoding of four soil samples from each plot, consisting of mineral and organic soil samples taken at approximately 3 m distance from the plot centre in each of the four cardinal directions. Plant species were recorded on a 200 m² quadratic area, distinguishing the herb (<1 m height) and shrub layer (>1–5 m height). We sampled arthropods with one Malaise trap, three pitfall traps, and one light trap per plot to cover different microhabitats and taxonomic groups. Arthropods from two out of three pitfall samples and moths from light traps were identified by taxonomists (see Table 1 for a list of identified taxonomic groups). Two pitfall traps adequately reflect forest plots similar to ours[80], but since single traps are sometimes disturbed (e.g., by wildlife), we placed three pitfall traps per plot. We then used only two out of the three pitfall samples to maintain comparability across plots per census. While beetles were identified for the entire sampling period, the other taxa from pitfall traps were only identified for the census around August (Table 1, Supplementary Fig. S5). Species from Malaise trap samples were identified through DNA metabarcoding for the entire sampling period. DNA metabarcoding amplifies DNA-fragments found in our samples – viable or not. However, this bias can be considered equal among all samples and does not affect the overall outcome. For an overview of the numbers of BINs/species of different taxonomic groups from DNA-metabarcoding and taxonomists, see Supplementary Fig. S7. Due to the different underlying species concepts, we analysed arthropods identified by taxonomists ("arthropods_{TAX}") and those identified by metabarcoding separately and refer to the latter as "insects_{BIN}", since we only analysed insects as they made up 96% of all arthropod BINs (Barcode Index Numbers) in Malaise traps. We recorded birds through passive acoustic recorders in the morning hours around sunrise and experts

**Table 1 | Descriptive summary of the sampling methods for each taxonomic group**

| Taxon | | Sampling method | Period | Number of sampling events/samples per plot | Mean No. sampling days/recordings per plot + range | Species identification |
|---|---|---|---|---|---|---|
| Bacteria | | Soil sampling | June 22th–Oct 26th | 1 | 1 | Metabarcoding |
| Fungi | | Soil sampling | June 22th–Oct 26th | 1 | 1 | Metabarcoding |
| Plants | | Vegetation sampling | May 22nd–Aug 8th | 1 | 1 | Expert |
| Arthropods$_{TAX}$ | Woodlice, Millipedes, Centipedes, Ants | Pitfall traps | July 20th–Sep 15th | 1 | 28.5 (23–34) | Expert |
| | Beetles | Pitfall traps | May 5th–Sep 15th | 3–5 | 105.3 (74–128) | Expert |
| | Moths | Light trap | June 15th–Aug 8th | 1 | 1 | Expert |
| Insects$_{BIN}$ | | Malaise trap | May 5th–Sep 15th | 2–7 | 105.3 (74–128) | Metabarcoding |
| Vertebrates | Birds | Audio recorder | Mar 25th - Aug 14th | 3–4 | 20.1 (9–21) | Expert |
| | Bats | Audio recorder | June 3rd–Sep 17th | 2 | 2 | AI + Expert |
| | Small mammals | Pitfall traps | May 5th–Sep 15th | 3–5 | 105.3 (74–128) | Expert |
| | Large mammals | Camera traps | May 4th–Oct 21st | 1–2 | 27.3 (14–28) | Expert |

While beetles and small mammals from pitfall traps were identified for the entire sampling period, the other taxa from pitfall traps were only identified for one census in August. The sampling period and thus the number of samplings for Malaise and pitfall traps varied due to shorter vegetation periods at higher elevations, as well as due to different pooling schemes of the Malaise trap samples from intensive monitoring plots. The number of censuses for birds and large mammals varies due to noise, failure of the device or the inability to reach plots because of snow cover. More information about the sampling and species identification procedures are provided in the supplementary methods for field sampling and metabarcoding. Supplementary Fig. S5 shows the dates of trap installation and samplings for each method.

identified species based on these recordings. Bats were recorded using ultrasonic recorders, identified to species level through automated software (batIdent, ecoObs, Nuremberg, Germany), and subsequently evaluated by an expert. Small mammals (i.e., mice, voles, dormice, and shrews) were caught in pitfall traps, and large mammals were recorded through wildlife cameras. Further details on the field sampling, DNA metabarcoding, and data preparation is provided in the supplementary methods section.

### Specialization measure
All calculations and analyses were conducted in R[81] (version 4.1.1). We used the reciprocal of the Simpson index to determine habitat diversity based on a species´ proportional use of forest developmental stages:

$$B = 1 / \sum_i p_i^2 \qquad (1)$$

where $p$ represents the number of occupied plots in stage $i$ relative to the total occupancies (proportional use) of a species. We calculated $B$ separately for each elevational zone to capture changes in a species´ habitat use with increasing elevation. We then used a null model approach which allows to account for the increasing probability of using more developmental stages with more occupancies (Supplementary Fig. S6). To simulate 500 random communities, we applied a non-sequential algorithm for presence/absence matrices which keeps matrix, row and column sums constant (function *permatfull* from vegan package[82]). We calculated a standardized effect size (SES) for each species and elevational zone by computing $B$ for each simulated community, subtracting the mean simulated from the observed $B$, and dividing by the standard deviation of the simulated $B$. As a result, we lost between four and 19% of species, depending on taxon and elevational zone, due to little variation and thus a standard deviation of zero (Supplementary Table S4). We multiplied the SES by -1 to obtain an index for specialization and averaged across species of each plot and elevational zone.

### Species richness and beta diversity
For taxa sampled through metabarcoding, DNA barcodes (bacteria: 16S, fungi: ITS, insects: CO1) were clustered into Operational Taxonomic Units (OTUs) using a similarity threshold of 98% for bacteria and fungi and 97% for insects. We used OTUs as a proxy for bacterial and fungal species, whereas insect OTUs were assigned to a globally unique identifier (Barcode Index Numbers, BINs) from the Barcode of Life data system (BOLD)[83,84], which we used as a proxy for insect$_{BIN}$ species. The utility of BINs in assessing biodiversity has been largely demonstrated[85–87]. We calculated species richness as the sum of all species found at a study plot. To quantify beta diversity, we calculated the Jaccard dissimilarity between each set of two plots from different forest developmental stages for each elevational zone, respectively.

### Statistics and reproducibility
We first analysed how the SES of niche breadth (H I), species richness (H II), and beta diversity (H III) changed with elevation (Fig. 2). In a second step, we analysed how differences in species richness between forest developmental stages differed between the three elevational zones (H IV). To do so, we fitted individual multilevel models for each taxonomic group within a Bayesian framework using the *brms* package[88]. We z-transformed (mean = 0, SD = 1) all continuous predictor variables to increase sampling efficiency.

To analyse habitat specialization along elevation, we used the SES of specialization as the response variable and the elevational zone as a three-level categorical predictor variable. Due to differences in the dispersion of the SES across the elevational zones, we fitted distributional models with a Gaussian error distribution and used the elevational zone to model the dispersion *phi*. We used multivariate Gaussian processes with x and y UTM coordinates as joint predictors to model spatial autocorrelation. We applied 15 basis functions for bacteria and 20 for all other taxa to minimize computation time and avoid overfitting but adequately account for spatial autocorrelation[89].

To address patterns of species richness along gradients of elevation and forest development, we fitted negative-binomial models with species richness as a response and an interaction term between forest developmental stage (categorical) and elevation (continuous) as predictors. For bacteria, fungi, and plants, we added the day of year as a continuous covariate to adjust for phenological effects due to different sampling dates. We added a variable that groups neighbouring study plots as random intercepts to account for spatial autocorrelation (groups are shown in Supplementary Fig. S1).

To address patterns of beta diversity among plots from different forest developmental stages of each elevational zone, we fitted models with the pairwise Jaccard dissimilarities as response and elevational zone as a categorical predictor. We added the elevational and spatial distance between each set of the two plots to all models, as well as the difference between the sampling day to the models for bacteria, fungi, and plants as continuous covariates, to adjust for spatial and phenological differences. In the case of a quadratic relationship between beta diversity and a covariate, we added the covariate as a quadratic term using the *poly* function. We fitted the models using a beta probability distribution.

We provide further information about model specifications (priors, iterations), evaluations (convergence, residual checks, goodness-of-fit (Supplementary Table S2), posterior predictive checks, VIF), and summaries (Supplementary Tables S5, S8a–S8c, S10, S12, and S13) in the statistical analyses and results of the supplementary material.

To assess the effect of elevation on habitat specialization (H I), we calculated the pairwise differences of the predictions between each elevational zone. To assess the overall effect of elevation on species richness (H II), we added the baseline elevation coefficient (gap stage) to all developmental stage and elevation interaction coefficients and averaged over all. To assess whether beta diversity decreases with increasing elevation (H III), we computed the differences of predictions between the elevational zones. To assess whether the differences among forest developmental stages are strongest at low elevations and decrease towards the tree line (H IV), we first computed the absolute differences between predictions of the optimum and each other developmental stage, separately for each elevational zone. We then computed a "difference-of-differences" across the elevational zones: We subtracted the absolute differences of each optimum-[developmental stage]-comparison of the montane and subalpine zone from the absolute difference of the corresponding optimum-[developmental stage]-comparison of the submontane zone. For a better understanding, we provide a graphical concept of our calculations in the supplemental material (Supplementary Fig. S6).

We summarize all posterior distributions using the Maximum A Posteriori (MAP) and the 95% Highest Density Interval (HDI), representing the value with the highest probability density (mode) and the interval containing 95% of the highest probability density (95% Credible Interval, CI). To assess the existence and importance of an effect, we calculated the probability of direction (pd) and the ROPE percentage for each predictor[90,91] using the bayestestR package[92]. The pd is strongly correlated with the frequentist p-value and represents a robust and model-independent index ranging from 50% to 100% that indicates the certainty of an effect's direction (positive or negative), i.e., the existence of an effect. However, the pd does not assess the magnitude or importance of an effect, which is better evaluated using the ROPE percentage. The Region of Practical Equivalence (ROPE) defines an area around the null value, enclosing values equivalent to the null and thus of negligible magnitude and importance. The ROPE percentage indicates the proportion of the 95% HDI within this area, which here is defined as a range from $-0.1 * SD_y$ to $0.1 * SD_y$[90]. All changes in species richness correspond to a unit increase in the standard deviation of elevation, which is 357 m across our plots. We provide all pd and ROPE percentage measures together with the model summaries in the supplementary results (Supplementary Tables S4-S12).

## Reporting summary
Further information on research design is available in the Nature Portfolio Reporting Summary linked to this article.

## Data availability
All data used in our analysis have been deposited in a publicly accessible archive on Dryad: https://doi.org/10.5061/dryad.bk3j9kdkp[93]

## Code availability
All analysis were conducted in R, and the code has been deposited in the same Dryad archive as the data: https://doi.org/10.5061/dryad.bk3j9kdkp[93]

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

## Acknowledgements

We thank Robin Reiter, Anna-Maria Bachleitner, August Schellmoser, Verena Styrnik, Ake Geiß, and a large number of colleagues from the National Park staff as well as student helpers for their contribution to field work. We further thank Alexander Szallies and Torben Kölkebeck (beetles), Wolfgang H. O. Dorow (ants), Jörg Spelda (woodlice, millipedes, centipedes), David Stille and Stefan Linzmaier (small mammals), and Alfred Haslberger (moths) for identification of species from pitfall and light traps. We thank Christoph Moning, Ralph Martin, Johannes Urban, Sascha Homburg, Lukas Griem, and Thomas Kuhn for bird species identification and Milenka Reiter-Mehr for bat species validations from sound recordings. We thank Lena Fleckenstein for DNA extraction and Torsten Hothorn for statistical consulting. This work was part of the "Climate Change Research Initiative of the Bavarian National Parks" funded by the Bavarian State Ministry of the Environment and Consumer Protection. Kristin Braziunas and Rupert Seidl acknowledge support from the European Research Council under the European Union's Horizon 2020 research and innovation programme (Grant Agreement 101001905, FORWARD).

## Author contributions

S.S. and R.S. designed the overall framework with inputs of C.B. and J.M. T.R. and S.S. developed the concept of the study with inputs from R.S. and J.M. L.G. and T.R. selected the study plots with help from S.S. and D.T. L.G. and T.R. collected the biodiversity data with help from S.S. and R.S. C.B. coordinated the laboratory work for DNA extraction. T.R. coordinated the taxonomic and L.G. and S.S. the molecular identification process. C.S. processed the LiDAR data. K.H.B. prepared the trait data for the plant species. L.G. and T.R. curated the data. T.R. analysed the data with support from S.S. and C.S. T.R. and S.S. led the writing with inputs from S.K.. All authors critically revised the manuscript and approved the final version.

## Funding

## Competing interests

The authors declare no competing interests.
