## [Transparent Peer Review file · Communications Biology]

Effects of climate and forest development on habitat specialization and biodiversity in Central European mountain forests

Corresponding Author: Mr Tobias Richter

Version 0:

Reviewer comments:

Reviewer #1

(Remarks to the Author)
Please find my review attached.

Reviewer #2

(Remarks to the Author)
The manuscript COMMSBIO-24-2061-T sheds light on the nuances of the Altitude-Niche-Breadth Hypothesis by assessing the individual and synergistic effects of climate and forest development on species richness and dissimilarity across multiple taxa along three elevational belts in the German Alps. A series of survey strategies allowed the authors to construct occurrence matrices for 150 forest plots equally representing 5 forest development stages. A Bayesian approach was used to model climatic and canopy cover variation among forest development stages and elevation gradients. With these inputs, the authors tested the hypothesis for groups of organisms at different taxonomic resolutions finding taxa-specific patterns. Based on these findings, the authors identify synergies between elevation and forest development and foresee mountains with richer arthropod communities and eroded plant and vertebrate communities in a future with a warmer climate.

General comments

The manuscript is clear in its goals and sufficiently detailed in the supplementary information. However, further clarifications within the main text, results, and discussion would highlight its contribution to the nuances of the altitudinal-niche-breadth-hypothesis. Regarding form, there are minor edits in the commented manuscript. Regarding content, it is admirable the omission of the intermediate disturbance hypothesis (IDH). Including this hypothesis and its criticisms (e.g., Fox, 2012, DOI:<https://doi.org/10.1016/j.tree.2012.08.014>) both as part of the theoretical background, in the introduction, as well as an explanatory framework for an important part of the findings, in the discussion session, would craft a more complete manuscript.

Specific comments on specific sections are detailed below. It is hard to review without numbered lines

Specific comments

Introduction:

1. Paragraph 2 needs a smoother transition to introduce the sentence "Climate is warming more strongly...".
2. See general comment about including IDH.

Results:

3. Start with stronger statements describing the main findings (like those provided in the first paragraph of the Discussion section, before "suggesting ...") rather than with numbers. Then provide details, including the numbers.
4. I missed actual number of B, the measure for niche breadth, for each group in the results.
5. As pd and ROPE are less common statistical parameters and methods are at the end of the manuscript, provide its definitions when supporting statements in the result section. Consider adding something like "...as indicated by the probability of direction (pd), a Bayesian measurement of evidence, and by ROPE, ..."

Discussion:

6. Rephrase and interpret the main findings in the first paragraph.
7. See general comments about including IDH.

Figures and Tables:

1. Figure 1. Panel IV does not seem to match panels I and II. Based on the first two hypotheses it would be expected higher

species richness in the highest elevational belt (subalpine) but this panel depicts submontane as the richer one. Clarify. 2. Figures 4 and 5. The reference number 66 repeats.

Supplementary material:

3. F1. Consider simplifying the bottom map removing graphical information presented in the top map (cover and forest development) and by representing plot groups either with the same icon or within boxes. As it is, the reader barely can distinguish colors among the small icons.

I look forward to reading this contribution published!

Version 1:

Reviewer comments:

Reviewer #2

(Remarks to the Author)

Manuscript COMMSBIO-24-2061A is the revised version of an assessment of the Altitude-Niche-Breadth Hypothesis in the German Alps. A series of survey strategies allowed the authors to construct occurrence matrices for 150 forest plots equally representing 5 forest development stages to assess the individual and synergistic effects of climate and forest development on species richness and dissimilarity across multiple taxa along three elevational belts in the German Alps. A Bayesian approach was used to model climatic and canopy cover variation among forest development stages and elevation gradients. With these inputs, the authors tested the hypothesis for groups of organisms at different taxonomic resolutions finding taxa-specific patterns. Based on these findings, the authors identify that "climate [...] and disturbance [...] independently drive patterns of species richness, but interactively shape community composition." With this evidence, the authors foresee higher differences in community composition in a warmer future.

General comments

The main manuscript and its supporting material are now clearer, as most of the comments from both reviewers have been addressed; the authors have provided good justification for not addressing some of the reviewers' comments. I have no major concerns about the content of this revised version.

I look forward to seeing this paper published!

Point-by-point response

Reviewer link to Dryad repository:

<http://datadryad.org/stash/share/ErFjLRpSfSnZNMtO9ZdYpQvOP23H26-9cIgv3bSxWkQ>

Reviewer #1 (Remarks to the Author):

The present study presents the results from an observational study in the German Alps along an elevational gradient of approx. 1100 m. Along this gradient, the authors established plots at three elevations (submontane, montane, subalpine) and five forest development stages (gap, establishment, optimum, plenter, terminal) with ten replicates for each combination, resulting in a total of 150 plots. At each plot, metabarcoding of soil was used to identify bacterial and fungal OTUs, and of malaise traps was used to identify insects. Additionally, plants, vertebrates and arthropods were identified by traditional methods. Using the data obtained, the authors investigate alpha and beta diversity, as well as niche breadth of “elevational” and “forest developmental” niches of the different groups. The authors conclude that the “altitudinal niche breadth hypothesis” is only supported for arthropods using their data. While the work presented here is certainly a contribution to the field, there are several points in this manuscript that need to be addressed. In total, I was surprised about the bold language when making inferences from this - at least regarding the elevation gradient – small study and the absence of a discussion of the shortcomings of the study.

Dear Dr. Schellenberger-Costa,

we thank you very much for your time and effort assessing our manuscript. We appreciate your critical but valuable remarks and suggestions, which we addressed by multiple changes throughout the manuscript. In particular, we now took care to adjust the wording to better reflect our local-scale study and replaced ‘niche breadth’ with ‘habitat specialisation’, which better fits the parameter we measured. We further added text regarding the problems of metabarcoding and clarified the discrepancies between species richness of taxonomic and barcoding data. Most importantly, we adjusted our calculation of habitat specialization (formerly niche breadth) based on your remark and calculate specialization based on the proportional use of developmental stages (the number of occupied plots per phase relative to the total occupied plots). Additionally, we now compute habitat specialization separately for each elevational zone to better capture intraspecific changes across the zones. Please find our detailed answers to all your comments below and please note that we only analysed how niche breadth/specialization with regard to forest developmental stages changed with elevation, but not elevational niche breadth/specialization.

My main points can be summarized as follows:

- given this is a local study, and others have found different patterns elsewhere, this study should make clear no general trends can be inferred, but just a local pattern

RESPONSE:

Thank you for this comment. We use a more careful wording to emphasize that our findings reflect local patterns. The title now includes “*Central European mountain forests*” and the abstract “*mountain forests of the Northern Alps*” (line 13). We further added “*in the Northern Alps*” in line 55, “*in Central European mountain forests*” in lines 145-147, 206-207 & 247-249, and “*in our study system*” in lines 252-255 & 277-280.

- the term “niche” is used in a very uncritical way without a proper definition here. I would argue that there is actually no “elevational” niche, because elevation is just a proxy for real niche dimensions (*sensu* Hutchinson). The same is true for the forest developmental stages. These are mixtures of many different niche dimensions (water availability,

radiation/temperature, ...). So, the authors need to define clearly how their concept of “niche” is to be understood (or avoid the term altogether and reframe the hypotheses). Be aware that your calculation of the “niche” is not enough – it is not what most people would understand a niche to be

RESPONSE:

Thank you for raising this point. First, please note that we did and do not calculate an elevational niche breadth (i.e., occurrence of species along the elevation gradient), but only with regard to forest developmental stages (i.e., occurrence of species along the forest developmental gradient) and analyse how the forest-development niche breadth change with elevation. Nevertheless, we fully agree that the distribution of species along the forest developmental gradient reflects many different niche dimensions (both abiotic and biotic variables). Measuring all potentially relevant niche dimensions/variables would have been impossible, especially across a multitude of taxa. Therefore, we used the forest developmental gradient as a comprehensive measure reflecting all drivers at play. The disadvantage of the approach is, however, that we cannot identify the contribution of the different niche dimensions in driving the observed patterns. We added a shorter comment on this to the Discussion (lines 260-266).

In addition, to avoid any confusion with Hutchinson's niche concept, we have replaced “niche breadth” with “habitat specialization” throughout the manuscript and use “habitat diversity” in the methods and supplement when describing our approach calculating habitat specialization and the SES (lines 359-361, supplementary Fig. S8). We revised our hypothesis to state that habitat specialization, based on a species’ proportional use of developmental stages, decreases with elevation (lines 59-62) and revised the whole manuscript accordingly.

- any hypothesis about changes in niche breadth along gradients should be tested against the full gradient. Picking just a part could work in theory, given the expectation of a linear decrease in niche breadth. However, the risk of getting any result because of the natural variability and detection uncertainties must be acknowledged and discussed. Additionally, having only three measurement points along the elevation gradient increases the risk of not capturing the actual pattern.

RESPONSE:

Thank you for the comment. Mountain areas offer the advantage of capturing significant environmental changes along elevational gradients over relatively short spatial distances. Our elevational gradient reaches the upper end at the tree line, but we acknowledge that it is truncated at the lower end and we agree that it would be favourable to study the full elevational gradient. However, elevation in our study region (and throughout most of the northern Alps) range between ~600 to 1800 m (with forest). Extending the gradient further downward would require extreme distances (it is approx. 1000 km to sea level in northern Germany). Having additional plots at lower elevation far away from the others would introduce bias from confounding factors (e.g., habitat fragmentation, no full forest development gradient in unmanaged forests, biogeography, and land-use history) which could substantially influence the results. But of course, even if is not feasible to sample a complete gradient, a truncated gradient could influence the results.

In our case, however, we are confident that the overall elevational patterns of specialization, alpha diversity and beta diversity would not change strongly when including the lowest elevational zone (< 600 m). The reason for this is that our gradient includes also the tree species and disturbance regimes which dominate the zone below 600 m. Specifically, our lowest elevational zone, the submontane zone, is already dominated by European beech which is also the dominant tree species of natural forest below 600 m. Moreover, the major changes in tree species composition along the elevational gradient, reflecting also major changes in abiotic conditions, happen within the range covered by our elevational gradient (i.e. submontane beech forest, montane beech-spruce-fir forest, subalpine spruce-larch-pine forest, Walentowski et al. 2020). Furthermore, following the arguments in Rasmann et al.

(2014), it is especially critical to include the upper part of the climatic gradient to test the *altitudinal-niche-breadth hypothesis*, which we did with our study. Testing hypotheses about changes in niche breadth/specialization along the full elevational gradient without spatial and other confounds, such as land use, would restrict studies to very specific areas and thus is a limitation inherent to almost all or maybe even all elevation-gradient studies worldwide. However, as we generally agree that hypothesis considering niche breadth (now habitat specialization) along gradients should be tested against the full gradient, we now include a remark concerning the interpretation of our findings in the caveats section (lines 266-275).

Concerning the additional comment regarding the three measurement points, this seems to be a partial misunderstanding. Although we selected our plots in a stratified way to cover the three elevational zones (= vegetation belts) equally, they were not all at the same elevation within one zone. These zones represent very distinct vegetation zones as outlined before, which is why we chose a stratified sampling design. Our statistical analyses of niche breadth and alpha diversity included elevation as a continuous variable, but to test if species richness differences and beta diversity between developmental stages change with elevation, we used elevational zones. In our revised approach, also due to your comments, we calculated habitat specialization of each species separately for each elevational zone. Therefore, and to capture potential non-linear patterns we now model specialization against elevational zones.

- The authors use metabarcoding for half of their data. Given that this study does capture DNA fragments, but no actual viable individuals/populations, a discussion on the problems of using this approach is necessary. The authors need to discuss the implications of having found 958 arthropod taxa with traditional methods, but DNA of 8335 taxa through metabarcoding.

RESPONSE:

We are sorry, we do not fully understand this comment.

- One possible interpretation of it is that the samples may contain DNA from individuals not directly trapped but carried by trapped organisms or fragments, such as hair etc, or e.g., dead microbial DNA. These DNA fragments could indeed represent non-viable organisms. We generally agree with the reviewer that this may affect our results for habitat associations, however, we see no reason to assume that this potential bias is not equal among all samples. Since we do not compare traditional methods with metabarcoding regarding absolute species/BINs/OTUs, this shortcoming does not affect the overall outcome. To clarify that we used DNA fragments, we now write that we use “DNA metabarcoding” for species identification of soil and malaise trap samples (lines 329 & 342) and we added text describing that DNA-fragments are being amplified – viable or not, but that we assume that this bias is equal among all samples and does not affect the overall outcome (lines 343-345). We further describe that DNA barcodes were clustered into OTUs, which we used as proxies for bacterial and fungal species; and that Malaise trap OTUs were assigned to a globally unique identifier (Barcode Index Number, BIN) based on the Barcode of Life data system (BOLD), which we used as a proxy for species and that the utility of BINs in assessing biodiversity has been largely demonstrated (Hausmann et al. 2013; Pentinsaari et al. 2014; Schmidt et al. 2015) (lines 377-383). We hope this adequately addresses the reviewers’ concerns.
- The discrepancy in the number of species/BINs between traditional methods and metabarcoding arises from two factors.
 1. For traditional methods, only beetles were identified for the entire sampling period, while other taxa (woodlice, millipedes, centipedes, and ants) were identified from a single four-week census in August and moths from a single night. In contrast, samplings of malaise traps used for metabarcoding encompassed the entire sampling period. This is shown in the descriptive

summary of the sampling methods (Tab. 1) as well as in the supplementary material for the field samplings. However, we acknowledge that we should better highlight this in the methods of the main body. We revised the sentence on identification methods (lines 340-343), which now reads: “While beetles were identified for the entire sampling period, the other taxa from pitfall traps were only identified for the census in August and light trapping took place on a single night per plot without rain and strong wind (Table 1, supplementary Fig. S5). Species from Malaise trap samples were identified through DNA metabarcoding for the entire sampling period.”

2. Malaise traps are known for their high sampling success and predominantly catch flying insects, but also ground-dwelling species, and since we used metabarcoding, all taxa were identified (Uhler et al. 2022). In contrast, light and pitfall traps are more selective for specific taxa and we only identified a subset of taxa (those that are adequately sampled by these trap types and for which taxonomists are available). Our Malaise trap samples were dominated by Diptera (50.2 %) and Hymenoptera (23.6 %), which accounted for almost 75 % of all BINs, followed by Lepidoptera (12.3 %) and Coleoptera (9.1 %) (supplementary Fig. S7), which is in line with other studies (Karlsson et al. 2020; Uhler et al. 2021). While Diptera and Hymenoptera are very species rich groups, the taxa identified using traditional methods mostly consist of species-poor groups. In a study across Bavaria, Uhler et al. (2021) collected over a shorter sampling period (Uhler: Mid-May – end of July, our study: End of April – start of September) a similar number of malaise trap samples (Uhler: 510, our study: 587) and identified 7589 BINs, which is quite similar to the 8335 BINs we had in our study. To give an overview of numbers of BINs/species of different taxonomic groups from DNA-metabarcoding and taxonomists, we prepared the supplementary Fig S7. and refer to the figure in the field sampling section in the main body (lines 345-347).

- a map showing all plots as well as absolute numbers of OTUs/species need to be provided
RESPONSE:

We thank the reviewer for this comment. Both were provided in the supplementary material and the absolute numbers are reported in the first paragraph of the results where we also refer to Table S3 (formerly Table S2) in the supplementary material (lines 69-73). Due to this comment, we decided to move the map to the main body (new Fig. 1).

- more comparisons to the literature would certainly benefit the discussion
RESPONSE:

Thank you for the comment. We added more literature to the discussion, e.g., Granot and Belmaker (2020) show a positive specialization-species richness correlation in a global study and suggest that specialization might be the outcome of the observed species richness (lines 172-176). We further added Peters et al. (2016) and (Brown 2014) linking species richness patterns with temperature (metabolic theory; lines 187-188) and Clavel et al. (2010) provides context regarding biological homogenization through loss of specialists (lines 215-217). We highlight the advantages and disadvantages of using forest developmental stages for forest research and management (lines 260-266: Franklin et al. 2002; Larrieu et al. 2014; Begehold et al. 2015; Hilmers et al. 2018; Hilmers et al. 2020) and we discuss the caveats of extending our truncated elevation gradient (lines 266-275: Beck et al. 2008; Gradstein et al. 2008; Haddad et al. 2015).

Additional comments

L13-14 This statement is far too general. At least add “in our study system”.

RESPONSE:

Thank you for the comment. We replaced it by "in the Northern Alps" (line 13).

L54 development on biodiversity

Changed accordingly

L55 covering all five stages

RESPONSE:

We think that including "five" would not be appropriate in this context as it implies that there are only five stages of forest development. Zenner et al. (2016) originally proposed nine stages and we adapted the protocol to five stages due to the similarity of some stages.

L59 development stages and elevation

Changed accordingly

L61 Why does species richness have to decrease when niche breadth increases? I do not see why it couldn't be otherwise.

RESPONSE:

Thank you for the comment. We acknowledge that "consequently" implies that niche breadth (now specialization) drives species richness, however, the process underlying the species richness-niche breadth pattern is still under debate (Carscadden et al. 2020). Furthermore, a recent global meta-analysis confirmed a negative correlation between community level niche breadth and species richness for many taxa and suggested that niche breadth is probably the outcome and not the cause of the observed species richness, at least along a latitudinal gradient (Granot and Belmaker 2020). We revised our first paragraph and refer to the negative niche breadth-species richness relationship (in the text positive specialization-species richness) correlation reported by Granot and Belmaker (2020) (lines 32-35: "Some studies provide evidence supporting the altitudinal-niche-breadth hypothesis and a recent meta-analysis confirmed a positive correlation between community-level specialization and species richness, yet the generality of these mechanisms across trophic levels and taxonomic groups remain inconclusive."), we revised our hypothesis I & II considering a positive specialization-species richness correlation (lines 59-62: "Based on the processes underlying the altitudinal-niche-breadth hypothesis and considering a positive specialization-species richness correlation, we predict that both (H I) habitat specialization and (H II) species richness decrease with increasing elevation."), and added to the discussion that specialization might be an outcome of species richness (lines 172-176: "Finally, specialization may not only drive species richness, but species richness could also affect specialization, as suggested by a recent meta-analysis on latitudinal specialization-richness patterns, which could be a further explanation for patterns (partly) inconsistent with the altitudinal-niche-breadth hypothesis.").

L76 Explain ROPE

RESPONSE:

We revised the text describing pd and ROPE as indices of effect existence and importance and referred to the methods (lines 82-87: "[...], we found strong evidence of higher habitat specialization in the submontane compared to the montane and subalpine zone for all taxa, as indicated by the probability of direction ($pd \geq 0.99$, a Bayesian measure of effect existence) and the ROPE percentage (0% in ROPE, a Bayesian measure of effect importance, supplementary Table S5; both measures described in more detail in the methods)."), where we explain both indices in more detail (lines 441-454).

L221 climate -> temperature

RESPONSE:

Thank you for this comment but given that besides temperature other climatic variables, such as precipitation, change with elevation, we do not agree with this change.

L241 between -> from, and -> to

Changed accordingly

L243 timberline -> tree line

Changed accordingly

L257 .. by planting of European ..

Changed accordingly

L274 Why aren't all plots shown?

RESPONSE:

The map was shown in the supplementary material but we moved it to the main body (new Fig. 1).

L284 Why two out of three?

RESPONSE:

Thank you for the question. Two pitfall traps adequately reflect forest plots similar to ours (Müller and Brandl 2009), but since single traps are sometimes disturbed (e.g. by wildlife), we placed three pitfall traps per plot. We then used only two out of the three pitfall samples to maintain comparability across plots per census. We revised the text accordingly (lines 337-339: "Two pitfall traps adequately reflect forest plots similar to ours⁷⁹, but since single traps are sometimes disturbed (e.g., by wildlife), we placed three pitfall traps per plot. We then used only two out of the three pitfall samples to maintain comparability across plots per census.").

L289 Explain BINs

RESPONSE:

We added the full term (Barcode Index Numbers) to the abbreviation (line 71) and give more detail about the preparation of OTUs and BINs and its use as proxies for species in lines 377-382: "For taxa sampled through metabarcoding, DNA barcodes (bacteria: 16S, fungi: ITS, insects: CO1) were clustered into Operational Taxonomic Units (OTUs) using a similarity threshold of 98% for bacteria and fungi and of 97% for insects. We used OTUs as a proxy for bacterial and fungal species, whereas insect OTUs were assigned to a globally unique identifier (Barcode Index Numbers, BINs) from the Barcode of Life data system (BOLD)^{82,83}, which we used as a proxy for insect_{BIN} species."

L303 ecological -> environmental

Changed accordingly

L308 It seems niche breadth as defined here is sensitive to the number of occurrences within replicates, i.e. a species occurring in all replicates has a larger niche breadth than a species occurring only once in each of the replicates, even if both cover the same forest types. This makes no sense to me, please explain.

RESPONSE:

Thank you for this important comment. We agree that there was a flaw in our niche breadth (now habitat specialization) calculation. We adjusted the formula to use the proportional use of

developmental stages (the number of occupied plots per phase relative to the total number of occupied plots) instead of the absolute use per stage. Additionally, we now compute habitat diversity and the SES separately for each elevational zone to better capture intraspecific changes across the zones. We then multiplied the SES of habitat diversity by -1 to obtain a measure for habitat specialization, averaged the SES across all species per plot within each elevational zone, and modelled it as a function of the elevational zones (three-level categorical variable). The new approach takes care of your comment and accounts for intraspecific changes in specialization along the elevation gradient. Based on this, we now find stronger patterns of decreasing specialization (increasing niche breadth) with elevation for all taxa. Please find the updated figure below (Fig. 3 in the main body).

Fig. 3 We updated the left panel based on our new results and replaced the former line with a violin graph and highlighted substantial differences between elevational zones with different point shapes.

L310 their proportion

Changed accordingly

L607/614 Repair double citations

Done accordingly

Fig 2 show data points

RESPONSE:

Thank you for the comment. Showing the data points allows to check data variability and model fit, but considering the number of points we would add to the figure, this would greatly impede readability due to overlaying points. Therefore, we provided a figure including data points in the supplement (Fig. S10), as noted in the legend of Fig. 3 (formerly Fig. 2). To better emphasize this, we revised the figure legend (lines 747-750) to indicate that the supplementary figures include data points: *"To show all taxa in a joint figure for species richness, we normalized the predictions of species richness to a range between zero and one based on each taxon's minimum and maximum values (see Table S3 and Fig. S9 in the supplement for values and figures including data points)."*

Fig 4 + 5 Given that Fig 4 is just the mean of Figs 5, it may be worth adding that mean per group in

Fig 5 and remove Fig 4

RESPONSE:

Thank you for this comment. We prefer to keep Fig. 4 as it intuitively illustrates the average effect across all six taxa in a single concise figure.

I hope this is helpful.

Yours sincerely,
David Schellenberger Costa

Reviewer #2 (Remarks to the Author):

The manuscript COMMSBIO-24-2061-T sheds light on the nuances of the Altitude-Niche-Breadth Hypothesis by assessing the individual and synergistic effects of climate and forest development on species richness and dissimilarity across multiple taxa along three elevational belts in the German Alps. A series of survey strategies allowed the authors to construct occurrence matrices for 150 forest plots equally representing 5 forest development stages. A Bayesian approach was used to model climatic and canopy cover variation among forest development stages and elevation gradients. With these inputs, the authors tested the hypothesis for groups of organisms at different taxonomic resolutions finding taxa-specific patterns. Based on these findings, the authors identify synergies between elevation and forest development and foresee mountains with richer arthropod communities and eroded plant and vertebrate communities in a future with a warmer climate.

Dear reviewer,

we would like to thank you for your time and effort assessing our study. We appreciate your valuable remarks and suggestions, which we addressed by several changes throughout the manuscript, helping to significantly improve its quality.

General comments

The manuscript is clear in its goals and sufficiently detailed in the supplementary information. However, further clarifications within the main text, results, and discussion would highlight its contribution to the nuances of the altitudinal-niche-breadth-hypothesis. Regarding form, there are minor edits in the commented manuscript. Regarding content, it is admirable the omission of the intermediate disturbance hypothesis (IDH). Including this hypothesis and its criticisms (e.g., Fox, 2012, DOI:<https://doi.org/10.1016/j.tree.2012.08.014>) both as part of the theoretical background, in the introduction, as well as an explanatory framework for an important part of the findings, in the discussion session, would craft a more complete manuscript.

RESPONSE:

Thank you for your suggestion regarding the inclusion of the intermediate disturbance hypothesis. The intermediate disturbance hypothesis is indeed an interesting hypothesis that has garnered a lot of insights into disturbance effects on ecosystems. However, our study design does not include a disturbance gradient. Consequently, our plots were not systematically selected based on disturbance severity, extent, or frequency, which means that we cannot meaningfully study biodiversity effects across varying disturbance levels. We here focused on forest developmental stages rather than different gradients of disturbance as our main factor in the analysis. As studying diversity effects of disturbance was not the objective of our study, and as our data are not well-suited to test the IDH, we have refrained from adding the IDH to the text as suggested by the Reviewer.

Specific comments on specific sections are detailed below. It is hard to review without numbered lines

We apologize for the missing line numbers; they were included in our submitted version.

Specific comments

Introduction:

1. Paragraph 2 needs a smoother transition to introduce the sentence "Climate is warming more strongly...".

RESPONSE:

Thank you for the comment. We added "Given that" to the start of the sentence and connected it with the next sentence to improve the transition (lines 47-51).

2. See general comment about including IDH.

See initial response about including IDH.

Results:

3. Start with stronger statements describing the main findings (like those provided in the first paragraph of the Discussion section, before “,suggesting ...”) rather than with numbers. Then provide details, including the numbers.

RESPONSE:

Thank you for the comment. These descriptive results aid in understanding the study system and, in our opinion, are best placed at the beginning of the results.

4. I missed actual number of B, the measure for niche breadth, for each group in the results.

RESPONSE:

Thank you for the comment. We now report B in the first paragraph of the results (lines 73-75) and added the average value of B and its range to the descriptive results shown in Table S3 in the supplementary material.

5. As pd and ROPE are less common statistical parameters and methods are at the end of the manuscript, provide its definitions when supporting statements in the result section. Consider adding something like “...as indicated by the probability of direction (pd), a Bayesian measurement of evidence, and by ROPE, ...”

RESPONSE:

Thank you for this comment and we totally agree. We revised the section accordingly and referred to the methods (lines 82-87: “[...], *we found strong evidence of higher habitat specialization in the submontane compared to the montane and subalpine zone for all taxa, as indicated by the probability of direction ($pd \geq 0.99$, a Bayesian measure of effect existence) and the ROPE percentage (0% in ROPE, a Bayesian measure of effect importance, supplementary Table S5; both measures described in more detail in the methods).*”), where we revised the text and explain both indices in more detail (lines 441-454).

Discussion:

6. Rephrase and interpret the main findings in the first paragraph.

RESPONSE:

We revised our interpretation of the findings based on our new results.

7. See general comments about including IDH.

See initial response about including IDH.

Figures and Tables:

1. Figure 1. Panel IV does not seem to match panels I and II. Based on the first two hypotheses it would be expected higher species richness in the highest elevational belt (subalpine) but this panel depicts submontane as the richer one. Clarify.

RESPONSE:

Thank you for the comment. Assuming that the x-axis increases from left to right and the y-axis increases from bottom to top, as is usual, panel II shows decreasing species richness with increasing elevation and therefore corresponds to panel IV.

2. Figures 4 and 5. The reference number 66 repeats.

Corrected.

Supplementary material:

3. F1. Consider simplifying the bottom map removing graphical information presented in the top map (cover and forest development) and by representing plot groups either with the same icon or within boxes. As it is, the reader barely can distinguish colors among the small icons.

RESPONSE:

Thank you for the comment. We decided to move the top map to the main body and we revised the bottom map that now only includes different point shapes representing the different plot groups.

I look forward to reading this contribution published!

Publication bibliography

Beck, Erwin; Bendix, Jörg; Kottke, Ingrid; Makeschin, Franz; Mosandl, Reinhard (2008): Gradients in a Tropical Mountain Ecosystem of Ecuador: Springer Science & Business Media.

Begehold, Heike; Rzanny, Michael; Flade, Martin (2015): Forest development phases as an integrating tool to describe habitat preferences of breeding birds in lowland beech forests. In *J Ornithol* 156 (1), pp. 19–29. DOI: 10.1007/s10336-014-1095-z.

Brown, James H. (2014): Why are there so many species in the tropics? In *Journal of Biogeography* 41 (1), pp. 8–22. DOI: 10.1111/jbi.12228.

Carscadden, Kelly A.; Emery, Nancy C.; Arnillas, Carlos A.; Cadotte, Marc W.; Afkhami, Michelle E.; Gravel, Dominique et al. (2020): Niche Breadth: Causes and Consequences for Ecology, Evolution, and Conservation. In *The Quarterly Review of Biology* 95 (3), pp. 179–214. DOI: 10.1086/710388.

Clavel, Joanne; Julliard, Romain; Devictor, Vincent (2010): Worldwide decline of specialist species: toward a global functional homogenization? In *Frontiers in Ecology and the Environment* 9 (4), pp. 222–228. DOI: 10.1890/080216.

Franklin, Jerry F.; Spies, Thomas A.; van Pelt, Robert; Carey, Andrew B.; Thornburgh, Dale A.; Berg, Dean Rae et al. (2002): Disturbances and structural development of natural forest ecosystems with silvicultural implications, using Douglas-fir forests as an example. In *Forest ecology and management* 155 (1-3), pp. 399–423. DOI: 10.1016/s0378-1127(01)00575-8.

Gradstein, S. Robbert; Homeier, Jürgen; Gansert, Dirk (2008): The tropical mountain forest. Göttingen: Göttingen University Press.

Granot, Itai; Belmaker, Jonathan (2020): Niche breadth and species richness: Correlation strength, scale and mechanisms. In *Global Ecol Biogeogr* 29 (1), pp. 159–170. DOI: 10.1111/geb.13011.

Haddad, Nick M.; Brudvig, Lars A.; Clobert, Jean; Davies, Kendi F.; Gonzalez, Andrew; Holt, Robert D. et al. (2015): Habitat fragmentation and its lasting impact on Earth's ecosystems. In *American Association for the Advancement of Science*, 2015. Available online at <https://www.science.org/doi/full/10.1126/sciadv.1500052>, checked on 9/5/2024.

Hausmann, Axel; Godfray, H. Charles J.; Huemer, Peter; Mutanen, Marko; Rougerie, Rodolphe; van Nieuwerkerken, Erik J. et al. (2013): Genetic patterns in European geometrid moths revealed by the Barcode Index Number (BIN) system. In *PloS one* 8 (12), e84518. DOI: 10.1371/journal.pone.0084518.

Hilmers, Torben; Biber, Peter; Knoke, Thomas; Pretzsch, Hans (2020): Assessing transformation scenarios from pure Norway spruce to mixed uneven-aged forests in mountain areas. In *Eur J Forest Res* 139 (4), pp. 567–584. DOI: 10.1007/s10342-020-01270-y.

Hilmers, Torben; Friess, Nicolas; Bässler, Claus; Heurich, Marco; Brandl, Roland; Pretzsch, Hans et al. (2018): Biodiversity along temperate forest succession. In *J Appl Ecol* 55 (6), pp. 2756–2766. DOI: 10.1111/1365-2664.13238.

Karlsson, Dave; Hartop, Emily; Forshage, Mattias; Jaschhof, Mathias; Ronquist, Fredrik (2020): The Swedish Malaise Trap Project: A 15 Year Retrospective on a Countrywide Insect Inventory. In *Biodiversity Data Journal* 8, e47255. DOI: 10.3897/BDJ.8.e47255.

Larrieu, L.; Cabanettes, A.; Gonin, P.; Lachat, T.; Paillet, Y.; Winter, S. et al. (2014): Deadwood and tree microhabitat dynamics in unharvested temperate mountain mixed forests: A life-cycle approach to biodiversity monitoring. In *Forest ecology and management* 334, pp. 163–173. DOI: 10.1016/j.foreco.2014.09.007.

Müller, Jörg; Brandl, Roland (2009): Assessing biodiversity by remote sensing in mountainous terrain: the potential of LiDAR to predict forest beetle assemblages. In *J Appl Ecol* 46 (4), pp. 897–905. DOI: 10.1111/j.1365-2664.2009.01677.x.

Pentinsaari, Mikko; Hebert, Paul D. N.; Mutanen, Marko (2014): Barcoding beetles: a regional survey of 1872 species reveals high identification success and unusually deep interspecific divergences. In *PLoS one* 9 (9), e108651. DOI: 10.1371/journal.pone.0108651.

Peters, Marcell K.; Hemp, Andreas; Appelhans, Tim; Behler, Christina; Classen, Alice; Detsch, Florian et al. (2016): Predictors of elevational biodiversity gradients change from single taxa to the multi-taxa community level. In *Nat Commun* 7 (1), p. 13736. DOI: 10.1038/ncomms13736.

Rasmann, Sergio; Alvarez, Nadir; Pellissier, Loïc (2014): The Altitudinal Niche-Breadth Hypothesis in Insect-Plant Interactions. In : *Annual Plant Reviews: John Wiley & Sons, Ltd*, pp. 339–359.

Schmidt, Stefan; Schmid-Egger, Christian; Morinière, Jérôme; Haszprunar, Gerhard; Hebert, Paul D. N. (2015): DNA barcoding largely supports 250 years of classical taxonomy: identifications for Central European bees (Hymenoptera, Apoidea partim). In *Molecular Ecology Resources* 15 (4), pp. 985–1000. DOI: 10.1111/1755-0998.12363.

Uhler, Johannes; Haase, Peter; Hoffmann, Lara; Hothorn, Torsten; Schmidl, Jürgen; Stoll, Stefan et al. (2022): A comparison of different Malaise trap types. In *Insect Conserv Diversity* 15 (6), pp. 666–672. DOI: 10.1111/icad.12604.

Uhler, Johannes; Redlich, Sarah; Zhang, Jie; Hothorn, Torsten; Tobisch, Cynthia; Ewald, Jörg et al. (2021): Relationship of insect biomass and richness with land use along a climate gradient. In *Nat Commun* 12 (1), p. 5946. DOI: 10.1038/s41467-021-26181-3.

Walentowski, Helge; Fischer, Anton; Kölling, Christian; Türk, Winfried; Rumpel, Alexander; Ewald, Jörg (2020): Handbuch der natürlichen Waldgesellschaften Bayerns. Ein auf geobotanischer Grundlage entwickelter Leitfaden für die Praxis in Forstwirtschaft und Naturschutz. 4. überarbeitete Auflage. Freising: Verlag Geobotanica.

Point-by-point response

Reviewer #1 (Remarks to the Author):

No remarks provided

Reviewer #2 (Remarks to the Author):

Manuscript COMMSBIO-24-2061A is the revised version of an assessment of the Altitude-Niche-Breadth Hypothesis in the German Alps. A series of survey strategies allowed the authors to construct occurrence matrices for 150 forest plots equally representing 5 forest development stages to assess the individual and synergistic effects of climate and forest development on species richness and dissimilarity across multiple taxa along three elevational belts in the German Alps. A Bayesian approach was used to model climatic and canopy cover variation among forest development stages and elevation gradients. With these inputs, the authors tested the hypothesis for groups of organisms at different taxonomic resolutions finding taxa-specific patterns. Based on these findings, the authors identify that “climate [...] and disturbance [...] independently drive patterns of species richness, but interactively shape community composition.” With this evidence, the authors foresee higher differences in community composition in a warmer future.

General comments

The main manuscript and its supporting material are now clearer, as most of the comments from both reviewers have been addressed; the authors have provided good justification for not addressing some of the reviewers' comments. I have no major concerns about the content of this revised version.

I look forward to seeing this paper published!

RESPONSE:

Dear reviewer,
we thank you very much for your time and effort assessing our manuscript a second time.